# Degenerative Disc Disease of the Spine: From Anatomy to Pathophysiology and Radiological Appearance, with Morphological and Functional Considerations

**DOI:** 10.3390/jpm12111810

**Published:** 2022-11-01

**Authors:** Luca Scarcia, Marco Pileggi, Arianna Camilli, Andrea Romi, Andrea Bartolo, Francesca Giubbolini, Iacopo Valente, Giuseppe Garignano, Francesco D’Argento, Alessandro Pedicelli, Andrea Maria Alexandre

**Affiliations:** 1Istituto di Radiologia, Università Cattolica del Sacro Cuore, 00168 Roma, Italy; 2Department of Neuroradiology, Neurocenter of Southern Switzerland-EOC, Via Tesserete 46, 6900 Lugano, Switzerland; 3Department of Diagnostic Radiology and Interventional Radiology and Neuroradiology, University of Pavia, IRCCS Policlinico San Matteo Foundation, 27100 Pavia, Italy; 4Diagnostic and Therapeutic Neuroradiology Unit, IRCCS INM Neuromed, 86077 Isernia, Italy; 5UOC Radiologia e Neuroradiologia, Dipartimento di Diagnostica per Immagini, Radioterapia Oncologica ed Ematologia, Fondazione Policlinico Universitario A. Gemelli IRCCS, 00168 Roma, Italy

**Keywords:** degenerative disc disease, disc degeneration, low back pain, discovertebral complex

## Abstract

Degenerative disc disease is a common manifestation in routine imaging of the spine; this finding is partly attributable to physiological aging and partly to a pathological condition, and sometimes this distinction is simply not clear. In this review, we start focusing on disc anatomy and pathophysiology and try to correlate them with radiological aspects. Furthermore, there is a special focus on degenerative disc disease terminology, and, finally, some considerations regarding disc morphology and its specific function, as well as the way in which these aspects change in degenerative disease. Radiologists, clinicians and spine surgeons should be familiar with these aspects since they have an impact on everyday clinical practice.

## 1. Introduction

Low back pain caused by degenerative disc disease (DDD) is a common manifestation that equally affects both sexes, from young to middle age with a median incidence usually around 40 years. If, on the one hand, it is demonstrated that disc degeneration increases with age, on the other, degenerated discs are not necessarily painful. It is plausible that standing is the evolutionary development that made the lumbar spine more vulnerable to degenerative disease. Degeneration is a common event to structures that compound the functional spinal unit: two adjacent vertebral bodies and the intervertebral disc. The disc and two zygapophyseal joints at the same level function as a tri-joint complex. As humans age, they undergo both macro- and microtraumas and endure changes that alter and redistribute biomechanical forces unevenly on the lumbar spine. Natural progression of degeneration of the lumbar segment of motion proceeds with characteristic anatomic, biomechanical, radiologic, and clinical findings in lumbar degenerative disc disease. The disc did not become the center of attention until 1933, when W.J. Mixter and J.S. Barr indicated the correct pathogenesis of lumbosciatic pain and nerve disfunction, and, hence, offered an appropriate surgical treatment [1]. The neurogenic nature of sciatica was first described by Domenico Cotugno (1764) [2]. Following him, outstanding French neurologists such as Lasegue, Dejerine, and Sicard enriched our understanding of the fundamentals of sciatica. The work of the German pathologists Schmorl and Andrea (1927–1929) established the modern basis for understanding the intervertebral disc by providing very clear discussions of herniations as well as degenerations [3,4]. This narrative review provides an overview from the literature about the processes and mechanisms of intervertebral disc degeneration, focused on the radiological ones. The scientific literature was reviewed through PubMed, Medline and Science Direct.

## 2. Anatomical Considerations: The Normal Disc and the Discovertebral Complex

The nucleus pulposus and the anulus fibrosus are the two main anatomical structures composing the intervertebral disc. Although its composition percentage varies in cervical, dorsal and lumbar spine, it is made of three basic components: proteoglycan, collagen, and water. The nucleus pulposus represents the inner part of the disc, and its high water content makes it suitable for tolerating high pressure either in standing or seated position [5]. The anulus fibrosus has a much more fibrous structure, containing a much higher collagen percentage and lower water content; several concentric sheets of collagen (also known as “lamellae”) allow the anulus fibrosus to function as a containing capsule of the nucleus because of the high pressure that it faces. The outer fibers of the disc, Sharpey’s fibers, have very tight connections with the vertebral plates and with the anterior and posterior longitudinal ligament. Both the nucleus and the anulus are avascular, thus depending on molecular diffusion from blood vessels at the disc’s periphery in terms of oxygen and nutrients supply and metabolic wastes removal [6]. This is a key concept in the anatomy and pathophysiology of the discovertebral complex, as it has been proved that a fall in nutrient supply can reduce the number of viable cells in the disc, leading to degeneration.

Histological composition of the lumbar intervertebral disc is similar to articular cartilage; chondrocyte-like cells have a pivotal role, generating type II collagen, proteoglycans, and non-collagenous proteins that collectively make up the matrix of the nucleus pulposus and the cartilaginous vertebral endplate. Fibroblast-like cells generate type I and type II collagen for the anulus fibrosus. Proteoglycans have a core protein structure with peripheral chains of glycosaminoglycans containing keratan sulphate and chondroitin sulphate. Proteoglycans in combination with hyaluronic acid chains form aggregates that are held together by type II collagen, which is cross-linked by type IX collagen.

The hydroscopic properties of the proteoglycan matrix provide hydrostatic properties to the nucleus, allowing it to accommodate compression loads and to brace the anulus. Chondrocytes also secrete other type of enzymes, called matrix metalloproteinases, that guarantee turn-over of the matrix constituents. Degradation of the matrix allows it to be refreshed by newly synthesized components. Matrix production is stimulated by several growth factors, such as basic fibroblast growth factor (bFGF), transforming growth factor (TGF) and insulin-like growth factor (IGF), which simultaneously inhibit the production of matrix metalloproteinases (MMPs) [7,8,9,10]. If the matrix is degraded, these growth factors that are normally bound by cartilage intermediate layer protein (CLIP) are released in order to promote further synthesis. Tissue inhibitors of metalloproteinases (TIMP) promote the same pathway, suppressing the activation of MMPs, thereby controlling degradation [11]. A decrease in pH reduces the rate of synthesis of matrix proteoglycans. Macrophages play a key role in the balance between production and degradation, inhibiting matrix synthesis and promoting MMPs activity through the production of cytokines such as interleukin-1 (IL-1), interferon (IFN), and tumour necrosis factor-α (TNF-α) [12]. TNF-α and IL-1 exacerbate inflammatory process [13]; they promote nitric oxide production stimulating inducible nitric oxide synthetase which has a variety of degradative effects. It affects matrix constituents directly, inhibits TIMPs, and thereby promotes matrix degradation and inhibits matrix synthesis. Macrophages also secrete superoxide (O2-) which is able to degrade hyaluronic acid and proteoglycans, causing their deaggregation, and can stop chondrocyte proliferation and synthesis [14].

The biochemical composition of the vertebral end-plates is close to that of the disc: water, proteoglycans, collagen and chondrocytes. The concentration scheme of these components follows the distribution observed in the disc: water and proteoglycan are the main components in the center of the end-plate, while, as we move toward the periphery, more and more collagen is present with less and less proteoglycans [6]. This similar biochemical distribution scheme helps the diffusion of nutrients between the subchondral bone of the vertebra and the depths of the disc.

## 3. Pathophysiology of Degenerative Disc Disease

The pathogenesis of degenerative disc disease is complex and multifactorial: many studies have found a strong genetic influence involving polymorphism in cytokine and matrix components genes [8,15,16,17]; risk factors involved are also aging, smoking, obesity, metabolic disorders, oxidative stress and low-grade infection [18,19,20,21,22,23]. Mechanical factors have also been traditionally linked to disc degeneration as it is more common and severe in the lower lumbar region where the mechanical stress is higher [16,17], though disc degeneration alone is not the cause of pain, as can be observed from the very high prevalence in the asymptomatic population. However, all the factors that we mentioned act in combination with inflammation that is a key factor involved in the degenerative process but also in the development of pain.

Many studies have demonstrated higher levels of cytokines such as IL-1, TNF-alpha, IL-6, IL-10, IL-4, IL-8 in the degenerated disc tissues [24,25] and in the serum of patients with degenerative disc disease [26]. The development of a pro-inflammatory microenvironment, both in the disc tissue and in the peri-discal space, leads to a gradual degradation of NP and AF matrix through the increased production of enzymes such as MMPs and ADAMTS (disintegrin and metalloprotease with thrombospondin-like repeats) [25,27]. These inflammatory mediators are produced by resident intervertebral disc cells and by circulating immune cells (e.g., macrophages, neutrophils, T cells, NK cells) which infiltrate the disc tissues in response to the production of chemokines by the intervertebral disc cells [25]; the migration is also permitted by the pathological disc matrix. Similarly, a reduction of the permeability and vascular supply of the cartilage endplates leads to changes in the microenvironment of the disc resulting in hypoxia, decreased nutrient supply and lower pH, and this is linked to an augmented production of pro-inflammatory mediators, matrix degrading enzymes, senescence and apoptosis of the resident intervertebral disc cells [28,29]. Cell apoptosis is increased in degenerated disc and the debris of the dead cells are not easily cleared since there is no vasculature and no resident immune cells, and this could be another trigger for inflammation [30]. Senescent cells stop replicating in response to mitogenic stimulation and aberrantly produce increased levels of cytokines and matrix-degrading enzymes [21,31,32].

The increased production of pro-inflammatory mediators in the disc tissues leads to dysregulation between synthesis and degradation of normal extra-cellular matrix components: in the degenerated disc, there is an increased production of collagen I and a decreased production of collagen II, an alteration in the distribution of collagen within the nucleus pulposus and anulus fibrosus and the progressive loss of proteoglycans in the nucleus pulposus, leading to the loss of hydration and turgor pressure in the disc [20], which, in turn, reduces the ability of the nucleus to resist compression [33] (Figure 1).

These changes lead to the loss of the physiological structure and mechanical function of the intervertebral disc [30] with loss of disc height and an increased risk of fissures formation in the anulus fibrosus with a possible disc herniation in response to various mechanical stimuli [33]. The degenerated disc can cause low back pain in several ways: one is the loss of disc height with successive modified stress on the facet joints, ligaments and muscles that can likely lead to back pain. Loss of disc height and disc herniation can contribute to mechanical compression of the exiting nerve root, with subsequent pain especially if the nerve is sensitized [34] (Figure 2).

Pain may even occur in the absence of disc herniation and root compression, as the degenerated disc itself can be the source of the pain: the discogenic pain is an important cause of chronic low back pain and is thought to be strictly related to the inflammatory cytokines such as TNF-alpha that may irritate nerve fibers and to the overexpression of substance P [25,35]. Furthermore, the growth of nociceptive nerves and vasculature into the usually (almost) aneural and avascular disc is a well-recognized feature of discogenic pain [25,35]. Nerve fibers from the sinuvertebral nerve penetrate much deeper, in the inner third of the anulus fibrosus, and also in the nucleus pulposus of the degenerated disc; furthermore, fibers are denser in the end plates of degenerated disc as compared to normal disc [36,37]. Nerve and vessel ingrowth is directly linked to the loss of structural and molecular integrity of the disc and is known to develop in the physically disrupted disc tissue [38] since the proteoglycans and other components of ECM physiologically provides an interstitial hydrostatic pressure which inhibits their growth in the normal disc [30,38,39]. Even the overexpression of growth factors such as NGF, BDNF [40] and VEGF is increased in the microenvironment of a degenerated disc and can contribute to the abnormal innervation and angiogenesis [27,30].

## 4. Radiological Considerations

Before MR imaging, the “gold standard” imaging modality to visualize spinal disc was the discography, which consists of the puncture of the intervertebral disc with a needle under fluoroscopy and successive injection of contrast agent [41]. Nowadays, MR is the gold standard for assessing the disc structure [42,43] and should be performed as the first examination. The most commonly used protocol for detection of degenerative disease of the spine includes sagittal T1- and T2-weighted images and sagittal short tau inversion recovery (STIR). The latter sequence helps in the characterization of edema of the medullary bone [44]. Axial T2-weighted images acquisitions are frequently performed on lumbar disc levels to detect potential foraminal or extraforaminal disc herniation (Figure 3).

Axial T2-Weighted images also help to identify the accurate relationship between the disc bulging (or herniation) and surrounding nerve roots and spinal cord. T2-Weighted sequences are very useful for characterizing the structure of lumbar intervertebral discs. For cervical spine exploration, a 3-dimensional gradient echo sequence provides a slice thickness of less than 1 mm, helping to see more precisely the intervertebral foramina and the nerve roots [41]. The administration of contrast agent should be reserved to well-selected patients, even if it can be very useful to study bone, intra-articular and muscular findings. Even though minimal signs of spondylolisthesis are noted in a standard supine MR, as, for example, excessive fluid in facet joints, bone instability must be suspected and a dynamic orthostatic X-ray or an orthostatic MR, if it is available, must be suggested to the patient. Specifically, the upright MR can be a supplementary diagnostic examination when there are negative results in conventional MR in symptomatic patients or when surgery is scheduled [45].

CT is not the method of choice for the study of the degenerative pathology of the spine, although it has some advantages compared with MRI, such as better visualization of osteophytes, intradiscal gas (vacuum disc) and calcifications. Alterations in discs are related to aging: desiccation, fibrosis and cleft formation in the nucleus, fissuring and degeneration of the anulus, sclerosis of end-plates or osteophytes at the vertebral apophyses are often seen in the asymptomatic patient during radiological examinations. During degeneration, type II collagen increases outside in the anulus and there is water decrease in the nucleus pulposus with resulting dehydration of the disc, known as desiccation. The disc becomes progressively fibrotic, with loss of distinction between anulus and nucleus. This results in uniform low signal on T2WI within the disc and often with loss of disc height [46]. Anular disruption is a known pathogenetic factor in degeneration of the intervertebral disc, and it occurs when there is a disjunction between the anular fibers and detachment from their vertebral body insertion. On imaging, this appears as linear high signal area on T2-weighted images (T2-w) within the anular fibrosis [46]. Disc degenerations should be classified with standard criteria using the Thompson Grading Scale [47], where, following a set of parameters, an X-ray radiographic inspection of the disc is conducted and the gross morphology is used to determine the extent of degeneration (Table 1).

It is a five-level grading scheme, ranging from grade 1 (not degenerated) to grade 5 (severely degenerated). This is a morphologic, and not a radiologic, classification, and therefore less useful for application in radiologic practice [47,48].

In 2001, Pfirrmann and colleagues [49] proposed a five-level grading scale for the evaluation of disc degeneration on T2-w degeneration (Table 2).

An even more useful approach for imaging studies is the modified Pfirrmann’s classification, which relies on mid-sagittal T2-weighted images to assess the intervertebral discs [48]. This system takes into account the signal intensity of the nucleus pulposus, the distinction between inner and outer fibers of the anulus fibrosus, and the height of the disc [50].

Unfortunately, common classifications of herniated disc disease are not generally accepted. Indeed, the term herniated disc means a focal lesion, and confusion with a bulging disc must be avoided. Radiologists are required anyway to specify the localization (median, paramedian, postero-lateral, lateral or foraminal and “far-lateral”), the fragment migration direction (cranial or caudal), and the continuity with the original disc.

Clinical features must be considered to determine if degenerative changes on imaging are pathologic and what may not have contributed to their development, even if distinguishing between aging changes and degenerative changes is not always easy. The role of imaging is to provide accurate morphologic information and influence therapeutic decision making. From the radiological point of view, a disc is considered “normal” when it is normally developed and free of any changes of disease, trauma, or aging. On the other hand, clinically “normal” means asymptomatic patient, even if in this case a wide variety of imaging findings can be present (congenital or developmental variations of discs, minor bulging of the anuli, age-related desiccation). Furthermore, the association between degenerative findings and pain should not be interpreted necessarily as causation [51]. Lumbar discs alterations represent only a relatively small part of degenerative spine pathology; in fact, eventual modifications of other spine structures such as ligaments, zygapophyseal joints, and vertebral body must be detected. These structures are involved in a mechanism cycle of degeneration by small changes in the mechanical integrity of the intervertebral disc. The intervertebral disc is the probable initial site of spinal degeneration and it is likely that the facet joint degenerates as a secondary result of disc degeneration [52]. Vertebral bodies can become displaced relative to the inferior level (process known as spondylolisthesis). This modification can lead to narrowing of the spinal canal and neural foramina [46]. In any case, compression effects on spinal canal, lateral recesses, neural foramina and their contents (thecal sac, nerve roots and ganglion) must be underlined if they are present. The axial T2-w images should be used to assess the central spinal canal, whereas the sagittal images should be used to assess the neural foramen [46]. Even dural venous plexus modifications must be searched for. If a central spinal canal stenosis is detected, the extension of the stenosis can be confirmed by the presence of varicose and thicker nerve roots above the site of compression.

## 5. Nomenclature

As previously stated, a fully and normally developed disc, free of changes in disease, trauma, or aging is a radiologically normal intervertebral disc. However, normal refers to an asymptomatic patient notwithstanding the presence of harmless radiological features (congenital or developmental variations of discs, minor bulging of the anuli, age-related desiccation, osteophytes). Starting from the Recommendations of the combined task forces of the North American Spine Society, the American Society of Spine Radiology and the American Society of Neuroradiology published by Fardon et al. in the version 2.0 [53], we have to clarify the terminology about degenerative disc disease and its classification:Aging disc: aging effects show loss of water content from the nucleus, which is an alteration that occurs before MRI changes consistent with the progressive loss of water content and the increase in collagen and aggregating proteoglycans [49].Disc degeneration refers to various alterations made in any of the following ways: desiccation, cleft formation, fibrosis, gaseous or mucinous degradation of the nucleus, fissuring, loss of integrity of the anulus, defects in and/or sclerosis of the end plates and to the presence of osteophytes at the vertebral apophyses. Imaging features of disc degeneration include morphological alterations such as disc space narrowing and peridiscal osteophytes. Various grading systems referring, respectively, to MRI disc modifications (Pfirrmann classification) [49], MRI changes of vertebral end plate, and subchondral bone marrow features (Modic classification) [16] can be used to assess those kinds of changes. (Table 3)Degenerative disc disease is a clinical condition evidenced by disc degeneration and symptoms associated with degenerative changes. A causal relationship between disc degeneration and symptoms such as low back pain is sometimes difficult to establish. However, the term “Degenerative disc disease” suggests an illness and the term should be considered as a nonstandard one when used instead of “degenerated disc” or “disc degeneration” for describing imaging features of the degenerative spine.Dark disc or black disc is a colloquial, nonstandard term used to describe a dehydrated disc. On MRI imaging, the disc loses its central high T2 signal intensity and it appears dark as a consequence of the dehydration of the nucleus.Disc height is defined as the distance between vertebral endplates on adjacent vertebrae. With degenerative disc disease, the intervertebral disc shrinks in height. In order to quantify the degree of such kind of modification, disc height should be measured at the center of the intervertebral disc, not at its periphery. If disc height is measured at the posterior or anterior margin of the disc on sagittal sections, this should be specified.Desiccated disc refers to a disc with reduced water content, predominantly of nuclear tissues. On MRI Imaging, the intervertebral disc shows decreased signal intensity on T2-weighted images (dark disc) as a consequence of the loss of water content and changes in the concentration of hydrophilic glycosaminoglycans.Vacuum disc refers to a degenerated intervertebral disc characterized by the presence of gas, predominantly nitrogen, within the disc space.Dallas classification [54] is a grading scale used to quantify the extent of anular fissuring seen on postdiscography CT imaging. According to the discogram description, grade 0 refers to a normal disc, grades 1, 2 and 3 describe a leakage of contrast into the inner one-third, the inner two-thirds and through the entire thickness of the anulus, respectively. In grade 4, the contrast media extends circumferentially; grade 5 is characterized by an overt contrast extravasation into the epidural space.Modic classification [16] is a classification for vertebral end plates and adjacent vertebral bodies MRI signal modifications secondary to disc inflammation and degenerative disc disease. Modic type 1 refers to decreased signal intensity on T1-weighted images and increased signal intensity on T2-weighted images. Such modifications may be chronic or acute and reflect the penetration of the end plate by fibrovascular tissue, inflammatory changes, and edema. Modic type II refers to increased signal intensity on T1-weighted images and isointense or increased signal intensity on T2-weighted images, indicating replacement of normal bone marrow by fat. Modic type III refers to decreased signal intensity on both T1- and T2-weighted images, indicating reactive osteosclerosis [55] (Figure 4).

## 6. Functional–Morphological Correlation

The intervertebral disc is a fibrocartilaginous junction that connects two adjacent vertebrae and is composed of three elements: a peripheral portion with a lamellar structure—“anulus fibrosus”—an internal portion—“nucleus pulposus”—and “the lamina cartilaginea” or vertebral plate, which covers the surfaces of the vertebral bodies [55]. The thickness of the fibrous ring in the anterior part of the disc, measured radially, is always greater than that of the posterior part, and the anterior peripheral lamellae are the most robust. This is congruent with the fact that the failure of the ring occurs preferably in the posterolateral region [56]. The stiffness of the layers is increasing from the center to the periphery, and this stiffness distribution is fundamental to explaining the collaboration of the various layers of fibers in the containment of the pulpy nucleus under pressure [57].

It is clear that damage to the external fibers is much more dangerous. Nutrition to the disc is dependent on the transfer of solutes from the vessels at the edges of the disc through the terminal vessels of the cartilaginous plate, in continuity with the systemic circulation via the lumbar arteries. Molecules move into and out of the disc by diffusion that depends on the concentration gradient and the nature of the solute molecules involved [58] (Figure 5).

This peculiar blood architecture influences both the supply of nutrients to the cells of the intervertebral disc and the clearance of their metabolites [59]. The main mechanical functions of the disc are to absorb the axial stresses and to resist the load applied to the spinal column by the force of gravity. The intervertebral disc is subjected to a variety of forces and moments [60]. Its main task is to absorb the compressive loads to which the spinal column is normally subjected; these loads are of three types: the weight of the overlying vertebra, the resultant of the muscle contraction forces necessary to maintain stability and execute the movements, and the weight of any objects lifted or transported. It is clear that the stresses acting on the disc are not constant, but vary in intensity and direction and are strictly connected to the activities carried out [61].

Due to its viscoelastic properties, the intervertebral disc behaves in a way resembling that of a flexible structure when subjected to limited loads, becoming more and more rigid as the load increases. These properties are an expression of the disc’s high adaptability to load forces. From a quantitative point of view, the disc distributes 75% of the load on the nucleus pulposus and 25% on the anulus [62].

A vertical pressure acting on the spine causes the compression of the disc with deformation of the nucleus pulposus which expands radially against the surrounding fibrous ring. The external expansion of the fibrous ring is necessary to absorb the forces to which the column is subjected [63]. They also remind us that in the absence of a load, the pressure in the center of the core is never zero. The core, in fact, is always in a state of pre-compression which gives the disc the ability to optimally resist compression forces, maintaining a certain flexibility at modest loads and becoming more and more rigid as the amount of the load increases [64]. Efficient functioning of the disc largely depends on the elasticity of the core in relation to its ability to retain water. The water is attracted by the oncotic pressure within the matrix of the nucleus pulposus, and is in dynamic equilibrium with the plasma [65]. The increase in intradiscal hydrostatic pressure favors the escape of water towards the plasma, while, on the contrary, the chemical composition of the pulpy nucleus attracts water inside [66].

In a lying position, when the forces are reduced, the water flows inside the disc; on the contrary, under load, the water is pushed out to compensate for the progressive increase in pressure. This process is faster the higher the load is. The cartilaginous plates are subject to less pressure and the stresses act more on the anulus, which can become damaged over time [67]. It has been known for some time that the height of the human body varies within 24 h. It can be assumed that, on average, an individual’s stature is shortened by about 1% during the period of daytime standing; in the elderly aged 70–80 the shortening is reduced to about 0.5% while it is 2% in young people [68].

Measurements of the resistance of the disc and its pressure show that the tensile force of the fibrous ring is between 15 and 50 kg/cm while that of the vertebral body varies between 8 and 10 kg/cm. The tension force that a disc can withstand in a healthy spine is 40 kg/cm [69]. When subjected to traction, the cartilage plat of the disc ruptures at a force of 850 N in the cervical region and 3000 N in the lumbar region. The vertebral bodies of these two zones, in compression, fracture, respectively, at 3000 and 5000 N [70]. The tension force of the longitudinal ligaments is about 200 kg/cm and offers good resistance to disc rupture. From these data, it can be understood why a vertebral body can be fractured without evidence of disc tear [71].

With a static load, the deformation of the disc depends on the duration of the force, becoming stable after about 5 min. The nucleus pulposus appears to act as an incompressible medium of short duration (1 s); subsequently, the nucleus and the anulus fibrous interact to redistribute and balance the weight to adapt to it [72]. This creep is a mechanism by which the disc can distribute stress. On the other hand, when a dynamic load is applied, the intervertebral disc can begin to vibrate, acting as a shock absorber and damping the oscillations [73]. The disc also possesses mechanical properties similar to those of many elastic systems [74]. For example, when a disc is subjected to a static load up to the elastic limit and has a further increase in dynamic load, the vibrations that intervene can, in the peak of intensity, exceeding the tensile limit of the anulus fibrosus, cause disc damage [75].

## 7. Conclusions

The intervertebral disc can be thought of as a soft pad that separates the vertebrae of the spine from one another. Its functions are mainly the following: it acts as a ligament by holding the vertebrae together; it acts as a shock absorber to carry axial load, and, finally, it acts as a pivot which allows spine bending, spine rotation and spine twisting. When functioning properly, the nucleus is a good example of what is called a “closed hydraulic system”; the restrained water of the nucleus (by the anulus and endplates) becomes incompressible to the axial load and transfers these forces to the lower vertebra. When this structure is damaged, then the “vicious circle” starts. Degenerative disease of the intervertebral disc is, in fact, still a common and remarkable health problem in the aged population, not completely understood, and it contributes significantly to the years of life disability. Multiple conditions are associated with intervertebral disc degeneration and discogenic pain, which makes a single definition limiting. In any case, precise diagnosis involves the combination of clinical and radiological findings.

## Figures and Tables

**Figure 1 jpm-12-01810-f001:**
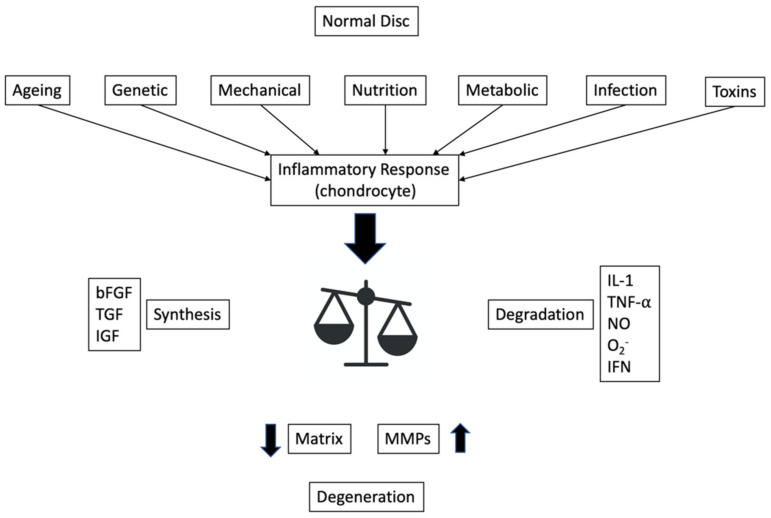
Schematic representation of inflammatory response and balance between degradation and synthesis cytokines leading to disc degeneration; modified from [17].

**Figure 2 jpm-12-01810-f002:**
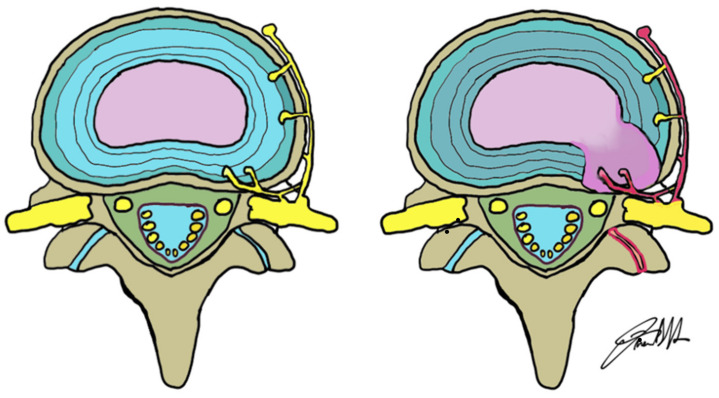
Schematic representation of the nerves surrounding the disc in a “normal” configuration on the left, and in case of a postero-lateral disc herniation (nerve root, red) on the right.

**Figure 3 jpm-12-01810-f003:**
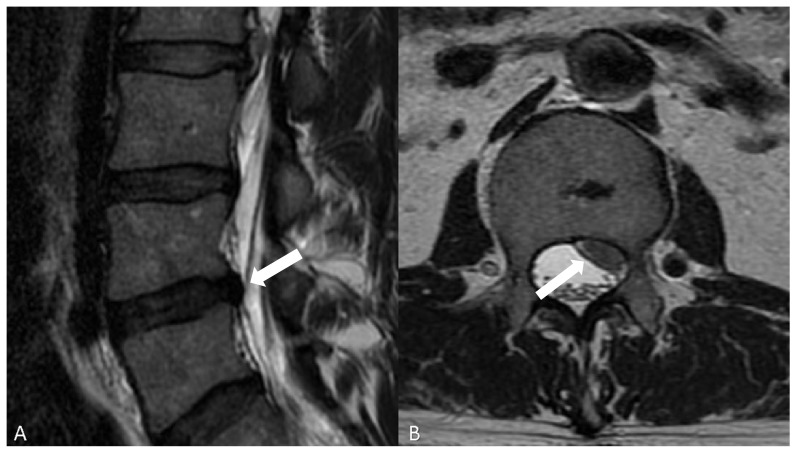
(**A**) T2-w, sagittal plan, showing multiple lumbar disc herniations, the most important at the level L3–L4 (white arrow); (**B**): T2-w, axial plan, showing a lumbar disc herniation, posterior, involving the left side of the spinal canal with compression of some nerves (white arrows).

**Figure 4 jpm-12-01810-f004:**
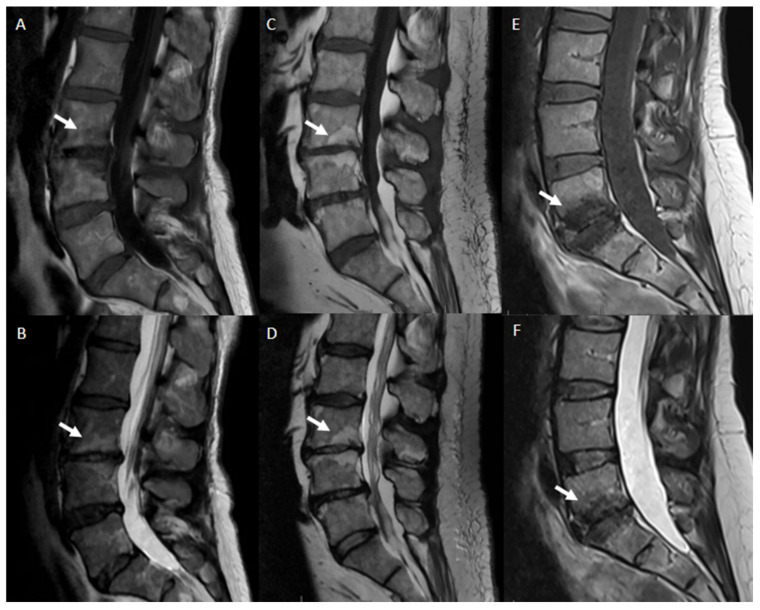
Modic alterations, sagittal T1-weighted and T2-weighted MR images. Modic 1 (**A**,**B**): T1-hypointensity and T2-hyperintensity at L3–L4 representing bone edema (white arrows). Modic 2 (**C**,**D**): T1-hyperintensity and T2-hyperintensity at L3–L4 representing fat degeneration and red (haemopoietic) bone marrow conversion to yellow (fatty) bone marrow(white arrows). Modic 3 (**E**,**F**): T1-hypointensity and T2-hypointensity at L5–S1 representing subchondral bone sclerosis (white arrows).

**Figure 5 jpm-12-01810-f005:**
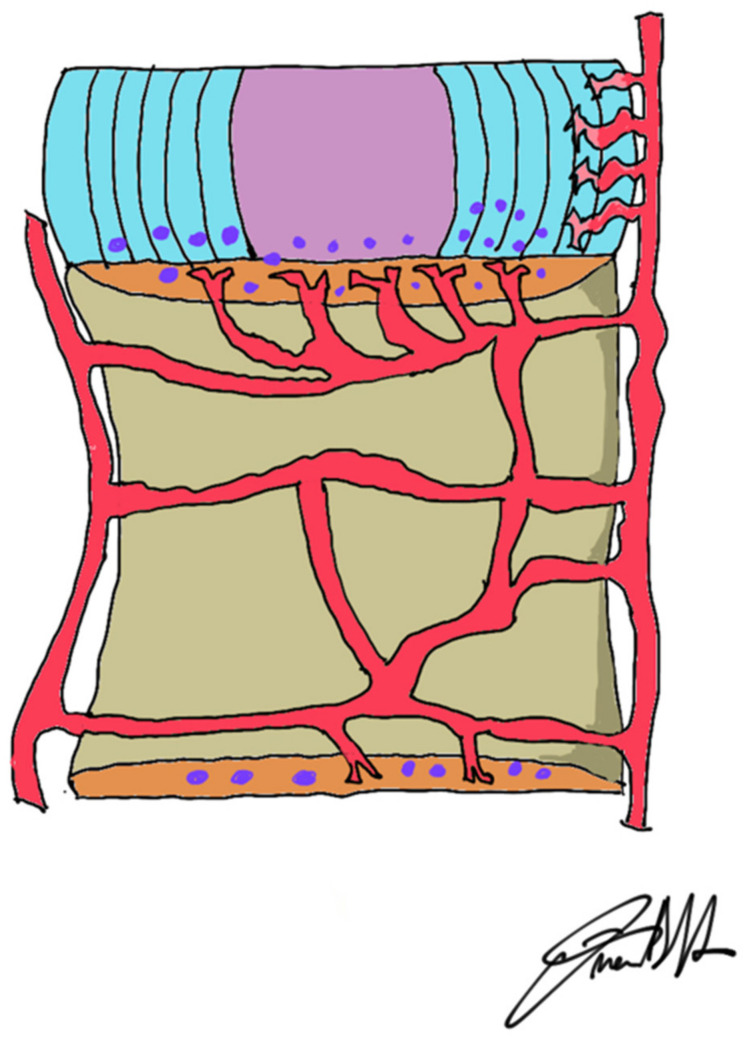
Nutrition of the disc depends mainly on diffusion of the nutrients through marrow cavities located in the vertebral end-plate. The nucleus and the inner two-thirds of the annulus are dependent upon simple gradient-related diffusion of molecules, while the outer anulus fibrosus receives nutrients directly from the capillary vessels.

**Table 1 jpm-12-01810-t001:** Thompson Grading Scale.

Grade	Nucleus	Anulus	Endplate	Vertebral Body
I	Bulging gel	Discrete fibrous lamellas	Hyaline, uniformly thick	Margins rounded
II	White fibrous tissue peripherally	Mucinous material between lamellas	Thickness irregular	Margins pointed
III	Consolidated fibrous tissue	Extensive mucinous infiltration; loss of anular demarcation	Focal defects in cartilage	Early chondrophytes or osteophytes at margins
IV	Horizontal clefts parallel to endplate	Focal disruptions	Fibro-cartilage extending from subchondral bone, irregularity and focal sclerosis in subchondral bone	Osteophytes less than 2 mm
V	Clefts extend through nucleus and annulus	-	Diffuse sclerosis	Osteophytes greater than 2 mm

**Table 2 jpm-12-01810-t002:** **Classification of Disc Degeneration by Pfirrmann**: grading scale used to assess the severity of degenerative changes within the nucleus of the disc. As the Pfirrmann classification states, grade I disc has a uniform high signal in the nucleus on T2-weighted images, grade II shows a central horizontal line of low signal intensity on sagittal images, grade III shows high intensity in the central part of the nucleus with lower intensity in the peripheral regions, grade IV shows low signal intensity centrally and blurring of the distinction between nucleus and anulus, and grade V shows homogeneous low signal with no distinction between nucleus and anulus.

Grade	Structure	Distinction of Nucleus and Anulus	Signal Intensity	Height of Intervertebral Disc
I	Homogeneous, bright white	Clear	T2-w Hyperintense, isointense to cerebrospinal fluid	Normal
II	Inhomogeneous with or without horizontal bands	Clear	T2-w Hyperintense, isointense to cerebrospinal fluid	Normal
III	Inhomogeneous, gray	Unclear	Intermediate	Normal to slightly decreased
IV	Inhomogeneous, gray to black	Lost	Intermediate to hypointense	Normal to moderately decreased
V	Inhomogeneous, black	Lost	hypointense	Collapsed disc space

**Table 3 jpm-12-01810-t003:** Classification of Modic changes.

Type	T1	T2	Histopathology
1	Hypointense	Hyperintense	Bone marrow edema
2	Hyperintense	Hypointense	Fatty replacement
3	Hypointense	Hypointense	Sclerosis

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
