# Peer review of "Degenerative Disc Disease of the Spine: From Anatomy to Pathophysiology and Radiological Appearance, with Morphological and Functional Considerations"

_jpm, 2022, doi:10.3390/jpm12111810_

Round 1

Reviewer 1 Report

I am honored to have the opportunity to review your manuscript. 

I thought that this manuscript is well described and organized with providing useful information about degenerative disc disease by composing various views on the anatomy, pathophysiology, radiology, and function. To be more clear and informative articles, I recommend to add the table and the representative MRI images depending on each grades of Modic changes.

Author Response

Thanks for your review

We have added the table and also some images of Modic changes

Reviewer 2 Report

The authors present a nice review of degenerative disc disease.  I appreciate their efforts in compiling the information.  

Author Response

Many thanks

Reviewer 3 Report

Thank you for your manuscript. This review is rich in content, describing the anatomy, pathophysiology and imaging evaluation of intervertebral disc degeneration, and listing some classic literature. The overall structure of this paper is complete, and the classical concepts in this field are reviewed. I think it can be revised and published. The main problem is that the description of clinical application evaluation is insufficient. What are the clinical effects of intervertebral disc degeneration at different stages and what are the guiding effects on treatment? The article would have been richer if related descriptions had been added. In addition, there are some details in the text, such as the abstract, degenerative should be Degenerative, In this review we start.....missing a ",".

Author Response

In our work, we mentioned the relationship between disc degeneration, 
the functional aspects and the treatment of the pathology, but this was not the goal of our review, which was basically focused on the radiological and pathophysiological aspects, as we wanted to emphasize already from the title.
Thank you anyway for the review, because it might be a good idea for a new narrative review given the extensive amount of studies in this area.
Thank you for the remaining comments, we have corrected the text.